# The Imbalance of Mitochondrial Fusion/Fission Drives High-Glucose-Induced Vascular Injury

**DOI:** 10.3390/biom11121779

**Published:** 2021-11-27

**Authors:** Yunsi Zheng, Anqi Luo, Xiaoquan Liu

**Affiliations:** School of Pharmacy, China Pharmaceutical University, Nanjing 211198, China; zys@stu.cpu.edu.cn (Y.Z.); luoanqi_cpu@163.com (A.L.)

**Keywords:** endothelial dysfunction, mitochondria dynamics, fusion/fission, metabolic memory, AMPK

## Abstract

Emerging evidence shows that mitochondria fusion/fission imbalance is related to the occurrence of hyperglycemia-induced vascular injury. To study the temporal dynamics of mitochondrial fusion and fission, we observed the alteration of mitochondrial fusion/fission proteins in a set of different high-glucose exposure durations, especially in the early stage of hyperglycemia. The in vitro results show that persistent cellular apoptosis and endothelial dysfunction can be induced rapidly within 12 hours’ high-glucose pre-incubation. Our results show that mitochondria maintain normal morphology and function within 4 hours’ high-glucose pre-incubation; with the extended high-glucose exposure, there is a transition to progressive fragmentation; once severe mitochondria fusion/fission imbalance occurs, persistent cellular apoptosis will develop. In vitro and in vivo results consistently suggest that mitochondrial fusion/fission homeostasis alterations trigger high-glucose-induced vascular injury. As the guardian of mitochondria, AMPK is suppressed in response to hyperglycemia, resulting in imbalanced mitochondrial fusion/fission, which can be reversed by AMPK stimulation. Our results suggest that mitochondrial fusion/fission’s staged homeostasis may be a predictive factor of diabetic cardiovascular complications.

## 1. Introduction

The incidence of diabetes is significantly increasing worldwide, with many people suffering from undiagnosed hyperglycemia [1]. Hyperglycemia is the most crucial risk factor for the development of diabetic vascular complications [2]. Despite restoration to normal glucose conditions, there remains a high risk of cardiovascular complications [3,4]. Accumulating evidence suggests that a prior history of hyperglycemia has persistent deleterious effects, as shown in both experimental models [5] and clinical trials [6,7], a phenomenon known as “metabolic memory”. Some mechanisms have been proposed for metabolic memory, such as oxidative stress, non-enzymatic glycation of cellular proteins, epigenetic changes, and chronic inflammation [8]. As mitochondria are the primary source of reactive oxygen species (ROS), these mechanisms point to mitochondria being the origin of the hyperglycemia-triggered vicious cycle.

Mitochondria are dynamic organelles that continuously undergo cycles of fusion and fission, which play a fundamental role in maintaining cellular function and survival [9]. Mitochondrial fusion is promoted by mitofusin 1 (Mfn1), mitofusin 2 (Mfn2), and optic atrophy 1 (Opa1), whereas fission is mainly controlled by dynamin-related protein 1 (Drp1) and fission protein 1 (Fis1) [10]. The harmonious tuning of mitochondrial fusion and fission balance is crucial for cellular adaptation to stimuli and stress. A common feature of mitochondria in a high-glucose environment is increased fragmentation, with up-regulation of Drp1 or down-regulation of Mfn1/2 [11,12]. Furthermore, tipping the balance towards increased fusion or decreased fission can ameliorate high-glucose-induced vascular injury [13]. Mitochondrial fusion and fission processes participate in the development and progression of diabetic cardiovascular complications [14].

The regulation mechanisms of mitochondrial fusion/fission balance shifting during diabetic progression have significant implications. One of the key modulators of mitochondrial fusion/fission balance may be the adenosine monophosphate-activated protein kinase (AMPK). AMPK is considered to be an energy switch controlling anabolism and catabolism [15]. Meanwhile, as a sensor of glucose and cellular energy status [16], AMPK is fundamentally needed to meet rapid mitochondrial fragmentation [17]. Previous studies have shown that AMPK activation can prevent drug-induced hepatocyte injury by promoting mitochondrial fusion [18] and also prevent diabetic endothelial apoptosis by inhibiting mitochondrial fission [19]. It has been known that phosphorylation of mitochondrial fission factor (MFF), the primary receptor for Drp1, by AMPK, is the underlying mechanism that mediates mitochondrial fission [20]. Recent evidence suggests that Mfn2 and Fis1 are regulated by AMPK during stress-induced autophagy and mitophagy [21,22].

Diabetic vascular disease is caused by endothelial dysfunction, which can reflect the severity of the vascular injury. Meanwhile, dysfunctional endothelial nitric oxide synthase (eNOS) in diabetic models is considered to play a critical role in the progression of vascular injury [23]. eNOS generates NO, which mediates vasodilation and resists arteriosclerosis [24]. The activity of eNOS is regulated by its phosphorylation levels. Previous work has demonstrated that the essential regulatory phosphorylated site of eNOS is Ser-1177 [25]. Hyperglycemia can induce vascular injury via suppressing eNOS phosphorylation at Ser-1177 [26].

Thus far, we do not have a clear understanding of the role of mitochondrial fusion/fission balance shifting in high-glucose-induced vascular injury. Herein, we investigated the dynamic alterations of mitochondrial fusion/fission during high-glucose-induced metabolic memory formation in vitro and in vivo. Our data suggest that mitochondrial fusion/fission’s staged homeostasis may be a predictive factor of diabetic cardiovascular complications.

## 2. Materials and Methods

### 2.1. Reagents and Chemicals

Dulbecco’s Modified Eagle’s Medium (DMEM), penicillin-streptomycin solution, 0.25% trypsin/EDTA, and phosphate-buffered saline were purchased from Gibco. Fetal bovine serum (FBS) was purchased from Biological Industries. Rabbit monoclonal anti-Mfn1(Cat#: 14739S), rabbit monoclonal anti-Mfn2(Cat#: 9482S), rabbit monoclonal anti-Opa1(Cat#: 80471S), rabbit polyclonal anti-Drp1(Phospho Ser616 Cat#: 4494S), rabbit monoclonal anti-AMPK_α_ (Phospho Thr172 Cat#: 2535S), and rabbit monoclonal anti-β-actin(Cat#: 4970T) were purchased from Cell Signaling Technology. Rabbit monoclonal anti-Fis1(Cat#: ab156865) and rabbit monoclonal anti-DRP1(Cat#: ab184247) were purchased from Abcam. DMSO, Mito-Tracker, BCA Protein Quantification Kit, RIPA lysis buffer, protease inhibitor cocktail, phosphatase inhibitors, SDS-PAGE Gel Preparation Kit, and other Western blotting reagents were purchased from Yeasen (Shanghai, China). Annexin V-Alexa Fluor 647/PI Apoptosis Detection Kit was purchased from Fcmacs (Nanjing, China). The mitochondrial membrane potential assay kit with JC-1 was purchased from Beyotime (Nanjing, China). Unless otherwise indicated, all other chemicals and reagents were purchased from Sigma-Aldrich.

### 2.2. Cell Culture and Experimental Designs

Human umbilical vein endothelial cells (HUVECs) were purchased from the American Type Culture Collection (Manassas, VA, USA), cultured in DMEM, supplemented with 100 U/mL penicillin-streptomycin with 10% FBS, and incubated at 37 °C with 5% CO_2_. Cells at 85–90% confluence from passages 4–20 were used for experiments. The cellular model of metabolic memory was built according to a previously described classic method [27], with some modifications. Briefly, HUVECs were exposed to a high-glucose medium (HG, 33 mM) for 0, 2, 4, 8, 12, 24, 48, 72, and 144 h, respectively, and then shifted to a normal-glucose medium (NG, 5.5 mM) respectively, all the cells were cultured for 144 h followed by harvesting. Mannitol was added to normalize osmolarity. Regardless of the type of media (NG or HG), the medium was changed every 24 h.

To explore the role of AMPK on high-glucose-induced mitochondrial fragmentation, NG group (negative control) cells were cultured for 144 h in the normal-glucose medium; HN group (model control) cells were cultured for 72 h in the high-glucose medium followed by 72 h in normal-glucose medium, and HG group (positive control) cells were cultured for 144 h in high-glucose medium. For ETC, AICAR, Met groups (AMPK agonists), cells were pre-incubated with a high-glucose medium for 72 h followed by a change to normal-glucose medium, containing 5 μM bempedoic acid (ETC), 0.2 mM AICAR, and 0.1 mM Metformin (Met) and cultured up to 144 h. For DOR, Compound C groups (AMPK inhibitors), cells were pre-incubated with normal-glucose medium for 72 h, then added 0.2 μM DOR or 1 μM Compound C, respectively, and cultured up to 144 h.

### 2.3. Western Blotting

After being washed twice with cold PBS, cells were collected and lysed in RIPA lysis buffer containing a protease inhibitor cocktail and phosphatase inhibitors. The cell lysates were homogenized by ultrasonication and centrifuged at 15,000× *g* for 5 min. The supernatant protein concentration was determined by using BCA Protein Quantification Kit. All the samples were diluted with 5× loading buffer and stored at −80 °C. Equal amounts of protein samples were resolved on 10% or 15% SDS-PAGE and transferred onto 0.45 μM PVDF membranes. The membranes were incubated with primary antibodies overnight at 4 °C after blocking for 2 h with 5% BSA in TBST, followed by HRP-conjugated secondary antibodies for 2 h at room temperature. The signals were visualized using Tanon ECL Western blotting detection reagents and the Tanon 5200 Gel Imaging System.

### 2.4. Apoptosis Assay

Apoptosis was determined by fluorescence-activated cell sorting (FACS) analysis (Accuri C6; BD Biosciences, San Jose, CA, USA) using double staining with Annexin V-Alexa Fluor 647 and propidium iodide (PI) according to the manufacturer’s instructions. Briefly, HUVECs were harvested with Accutase, resuspended in 0.1 mL 1× binding buffer, incubated with 10 μL Annexin V-Alexa Fluor 647 for 15 min at room temperature in the dark, then washed with 1 mL 1× binding buffer and centrifuged at 300× *g* for 5 min, resuspended with 0.5 mL 1× binding buffer, and incubated with 5 μL PI for 5 min in the dark, and then subjected to flow cytometry analysis within 1 h.

### 2.5. Measurement of Mitochondrial Membrane Potential

According to the manufacturer’s instructions, the mitochondrial membrane potential was monitored by fluorescence-activated cell sorting (FACS) analysis (Accuri C6; BD Biosciences, San Jose, CA, USA) using JC-1 dye. Briefly, the HUVECs suspension was incubated with JC-1 working solution at 37 °C for 20 min, centrifuged at 600× *g* for 5 min, washed with 1 mL JC-1 buffering solution twice, and resuspended in 0.5 mL JC-1 buffering solution. The FITC channel monitored JC-1 monomers (the green fluorescence), and the PI channel monitored JC-1 aggregates (the red fluorescence). The mitochondrial membrane potential was assessed by the red/green fluorescence intensity ratio, and all the values were normalized to the NG group.

### 2.6. Animal Models and Treatments

The apoE KO mice (C57BL/6 background, 7 weeks old) were purchased from the Sipol Bikai Lab Animal Co., Ltd. (Shanghai, China) for this study. Our research followed the Guide for the Care and Use of Laboratory Animals (NIH publication, 8th edition, 2011). This metabolic memory model came from a classic method [28]. Briefly, 60 male mice received 5 daily intraperitoneal injections of 55 mg/kg/day streptozocin (STZ). Meanwhile, 6 male mice received vehicle injections (0.1 mol/L citrate-phosphate buffer, PH 4.5) as control. All mice were fed for 20 weeks. We measured the fasting blood glucose levels every Monday afternoon. Despite initially developing hyperglycemia, about 20% of the STZ-injected mice had reduced blood glucose levels (<16.7 mmol/L), which remained decreased until the time of sacrifice. Based on the duration of hyperglycemia exposure and vascular injury, mice were divided into four groups: the Veh (Vehicle) group (with no hyperglycemia), the HG group (with persistent hyperglycemia), the HN-A group (transient hyperglycemia without vascular injury), and the HN-B group (transient hyperglycemia with vascular injury). All these mice received standard food and water ad libitum and fasted for 8 h before measuring blood glucose. At the end of the experiments, aortic root and abdominal aortas were collected.

### 2.7. Plaque Area Quantitation

Aortic root atherosclerotic lesions were analyzed by Oil Red staining [29]. The plaque area was quantified as the percentage area of the aorta stained red (CaseViewer, 3DHISTECH, Sysmex, Switzerland).

### 2.8. Immunohistochemistry Assay

Immunohistochemistry staining was performed to detect Mfn1, Mfn2, Opa1, Fis1, p-Drp1 (Ser616), p-eNOS (Ser1177), and p-AMPK (Thr172) in endothelial cells in mouse abdominal aortas. The semiquantitative analysis of tissue staining was performed using an arbitrary grading system from the product of the staining intensity (score 0: unstained; score 1: faint yellow; score 2: pale brown; score 3: brown) and the percentage of positive cells (score 0: 0–5%; score 1: 6–25%; score 2: 26–50%; score 3: 51–75%; score 4: 76–100%).

### 2.9. Mitochondrial Morphology Imaging

The images were acquired using the Opera Phenix High Content Screening System (PerkinElmer, Waltham, MA, USA). The fluorophores were detected with the following excitation and emission (Ex/Em) wavelengths: Mito-Tracker (644/665) and Hoechst 33342 (405/435-480). Mitochondria fragmentation and quantification analysis were performed using the Harmony software version 4.9. For analysis of the ratio of fusion, the following steps were carried out: the nucleus and mitochondrial skeleton were identified; the mitochondria and cells were mapped; the mitochondrial length of the normal-glucose group was adjusted so that the fragmented mitochondria accounted for about 20%; this mitochondrial length (3 nm) was set as the standard to distinguish fusion and fission, the fusion ratios of other groups were obtained.

### 2.10. Statistical Analysis

The results were presented as mean ± SEM, and all data were analyzed using GraphPad Prism software version 6.0 (SanDiego, CA, USA). One-way analysis of variance (ANOVA) followed by Fisher’s LSD post hoc test was applied to determine the statistical differences among the groups. *p* < 0.05 was considered to be statistically significant. All experiments were repeated at least three times to ensure reproducibility.

## 3. Results

### 3.1. Progression of Metabolic Memory in High-Glucose-Exposed HUVECs

To study the progression of high-glucose-induced vascular injury, we used HUVECs with different high-glucose pre-incubation as a model. Following 8 h of high-glucose exposure, the mitochondrial membrane potential (MMP) significantly decreased gradually (Figure 1c). After 12 h of high-glucose exposure, the apoptosis ratio showed a significant rise (Figure 1b). Meanwhile, the eNOS phosphorylation at serine 1177 declined (Figure 1d,e). These data suggest that vascular injury can develop very soon after high-glucose exposure.

### 3.2. The Staged Mitochondrial Fusion/Fission Homeostasis during Metabolic Memory Formation

To examine whether the high-glucose-induced vascular injury was accompanied by shifts in mitochondrial fusion/fission balance, we assayed the alteration of mitochondrial fusion/fission proteins expression (Figure 2a–f). Mfn1, Mfn2, p-Drp1/Drp1, and Fis1 showed a marked change after 2 h of high-glucose exposure, whereas Opa1 decreased after 24 h high-glucose exposure. It is worth noting that the expression of Fis1 increased initially and then decreased along with metabolic memory formation. To better understand the profile of mitochondrial fusion/fission balance shifting, we represented our results as an annotation graph (Figure 2g). The mitochondrial fusion/fission balance alteration could be divided into the compensation stage, balance shifting stage, and decompensation stage. At the initial stage, mitochondria can maintain a normal MMP despite perturbations in the balance of fusion/fission. Then, with the extended high-glucose exposure, there is a transition to the mitochondria progressive fragmentation due to the imbalanced mitochondrial fusion/fission; meanwhile, the MMP is reduced, but no apoptosis was observed during this stage. Finally, with suppressed Fis1 expression, mitochondria fragmentation triggers cellular apoptosis.

To examine whether the alteration of mitochondrial morphology coincided with the changes of mitochondrial fusion and fission proteins, we performed mitochondrial morphology visualization at the 144 h endpoint for all glucose stimulation groups (Figure 3). With the extension of high-glucose exposure, mitochondria gradually became fragmentary, which was consistent with the changes of mitochondrial fusion and fission proteins.

### 3.3. In Vivo Verification of the Staged Balancing of Mitochondrial Fusion/Fission

To confirm the staged balancing of mitochondrial fusion/fission during metabolic memory formation in vivo, we recreated a classic model of diabetic cardiovascular disease [28]. This model has been used to study metabolic memory in several papers [30,31,32]. The mouse has such an efficient lipoprotein clearance system that only genetic modification can accomplish this atherogenic profile or at least one that allows vascular disease within an acceptable period. Deletion of apoE, the protein involved in receptor clearance of remnant lipoproteins created from VLDL and chylomicrons, is a widely used recipe for creating accelerated atherosclerosis mice. STZ (Streptozotocin) selectively destroys the islet cells and induces hyperglycemia. According to the Animal Models of Diabetic Complications Consortium guideline, we performed multiple low doses of STZ to minimize the nonspecific toxic effects of STZ and provide a robust and consistent hyperglycemic response.

Based on the duration of hyperglycemia exposure (Figure 4a) and vascular injury (Figure 4b), mice were divided into four groups: the Veh (Vehicle) group (with no hyperglycemia), the HG group (with persistent hyperglycemia), the HN-A group (transient hyperglycemia without vascular injury), and the HN-B group (transient hyperglycemia with vascular injury). Compared with the Veh group, the HN-B group and the HG group developed much greater atherosclerotic lesions at the aortic root, accompanied by suppressed eNOS phosphorylation levels on abdominal aortic sections (Figure 4b–e). Accordingly, there was no apparent vascular damage in the HN-A group. Thus, 6 weeks’ exposure to hyperglycemia was a critical branching point in the progression of high-glucose-induced vascular injury. 

The early-stage (the HN-A group) showed decreased Opa1 expression and increased Drp1 phosphorylation; in the advanced stage, decreased Mfn1/2 and Fis1 were also observed in the HN-B group (Figure 5). These results prove that metabolic memory progression is accompanied by staged temporal alterations in mitochondrial fusion/fission balance.

### 3.4. Involvement of AMPK in High-Glucose-Induced Mitochondrial Fusion/Fission Balance Shifting

To explore the role of AMPK on high-glucose-induced mitochondrial fragmentation, we used the phosphorylation of AMPK at Thr172 as an indicator of AMPK activity. Our results show that AMPK was inhibited during high-glucose-induced vascular injury both in vitro and in vivo (Figure 6). Moreover, AMPK was mildly inhibited in HN-A but more significantly in HN-B, suggesting that AMPK was involved in high-glucose-induced vascular injury. The AMPK agonist bempedoic acid (ETC) and the AMPK inhibitor dorsomorphin dihydrochloride (DOR) were used to mimic the effects of AMPK.

The effects of some other classical AMPK activators and inhibitors were also evaluated (Figure 7a,b). DOR and compound C exhibited similar inhibition effects, whereas ETC, AICAR, and metformin (Met) showed similar activation effects. ETC restored AMPK activity and mitochondrial membrane potential, and significantly increased the expression of Mfn1 and Mfn2, and restored Opa1, Fis1, and Drp1 phosphorylation levels. Whereas DOR can mimic high glucoses’ effects on AMPK phosphorylation, mitochondrial membrane potential, mitochondrial morphology, Drp1 phosphorylation at Ser616 and Fis1 expression (Figure 7c–g). Together, these findings suggest that AMPK plays a vital role during mitochondrial fusion/fission balance shifting.

## 4. Discussion

In the present study, we found that hyperglycemia-induced vascular injury could develop within a short time. During this process, mitochondrial fusion/fission balance exhibits a staged shifting. Fis1 could be a promising predictor for the mitochondrial fusion/fission balance shifting during the high-glucose-induced vascular injury. Furthermore, AMPK mediated mitochondrial fusion/fission balancing during the development of hyperglycemia-induced vascular injury.

Despite intensive control of hyperglycemia, there remains a high risk of death from diabetic cardiovascular complications [8]. Our findings suggest that transient high glucose exposure leads to persistent vascular injury even if glucose is well controlled. Further investigations should pay more attention to the early stage of hyperglycemia-induced vascular injury. As general regulators of energy, mitochondria play a crucial role in the adaption to environmental stimuli. Recent studies have found the fragmentation of mitochondria in multiple cells of diabetic patients, such as hepatocytes [33], skeletal muscle cells [34], beta cells [35], adipocytes [36], and endothelial cells [37]. Mitochondrial fragmentation is considered a “starting point”, which triggers oxidative stress and cellular apoptosis in diabetic vascular complications [13]. Mitochondrial fission does not necessarily result in complete loss of mitochondrial function [38], and our results also support this viewpoint. With the extension of high-glucose exposure, mitochondrial morphology was inclined to fragmentation, whereas mitochondrial fusion/fission balance undergoes three stages: the compensation stage, balance shifting stage, and decompensation stage. At the compensation and balance shifting stages, persistent vascular damage had not developed. In the decompensation stage, mitochondria fragmentation occurred before persistent vascular damage, which suggests high glucose may trigger the pathologies of diabetic vascular complications via the mitochondrial fusion/fission imbalance.

We wondered if the mitochondrial fusion/fission balance shifts toward fission or not to fusion during high-glucose-induced mitochondrial fragmentation. Our results show that fusion proteins Mfn1, Mfn2, and Opa1 were rapidly suppressed by high glucose in HUVECs, whereas only Opa1 was suppressed in transient hyperglycemic mice (the HN-A group), which suggests the importance of the inner mitochondrial membrane fusion regulated by Opa1. Opa1 is cleaved from the long opa1 (L-Opa1) to the short Opa1 (S-Opa1) form, and this process has been highlighted as a critical step in coordinating mitochondrial fusion and fission [39]. The fission activity of Drp1 is controlled by various post-translational modifications, such as phosphorylation. The phosphorylation in human Drp1 on Ser616 and Ser637 residues is studied extensively. Previous work indicates that hyperglycemia induces an increase in mitochondrial fission via phosphorylation of Drp1 at Ser637 [40]. However, a recent study confirmed that the phosphorylation of Ser637 is not mainly related to mitochondrial fission. The phosphorylation of Ser616 plays a more critical role in fragmenting mitochondria [41]. A previous study has found that both Drp1-mediated fission and Opa1-mediated fusion require a certain level of MMP [42], which suggests that MMP contributes to the mitochondrial dynamic balance. In our study, MMP gradually decreased during the balance shifting stage; meanwhile, Fis1 began to decline, and Drp1 remained unchanged in vitro. As a receptor for Drp1 recruitment, Fis1 exhibited inhibitory effects toward fusion, whereas Drp1 is inconsequential for Fis1-mediated mitochondrial fragmentation [43]. Collectively, our study provides evidence that fusion inhibition plays a dominant role in high-glucose-induced mitochondrial fragmentation instead of fission activation.

On the other hand, our results suggest that Fis1 could be a promising marker for high-glucose-induced vascular injury. Shenouda et al. reported that Fis1 was significantly increased after 24 h incubation with 30 mM glucose in endothelial cells [11]. However, Wang et al. showed that Fis1 was not affected by 24 h incubation of 30 mM glucose in HUVECs [44]. This controversy could be attributed to the lack of overall observation of mitochondrial fusion/fission dynamics. In the present study, high glucose initially increases in Fis1 followed by decreases both in vitro and in vivo, suggesting that mitochondrial fusion/fission continuously undergoes dynamic alteration. Our experiment provides information about the influence of long-lasting hyperglycemia on mitochondrial dynamics, which allows a fuller understanding of the effects of mitochondrial homeostasis on diabetic vasculopathy. The previous report showed an apparent correlation of mitochondrial function with precise Fis1 expression levels [35]. Furthermore, dynamic Fis1 loss results in aberrant Syntaxin 17 accumulation on mitochondria, which triggers a PINK/Parkin-independent mitophagy [45]. According to these studies and our results, we propose that the progression of high-glucose-induced vascular injury depends on precise Fis1 expression.

As a glucose and energy sensor, AMPK regulates the processes of both fusion and fission [15,21,22,36,46,47]. Meanwhile, AMPK has been reported to be involved in the development of diabetic vasculopathy [19]. This evidence indicates that AMPK signaling could be an underlying link that connects high-glucose stress and mitochondrial fusion/fission balance shifting. Previous work has found that the hierarchical activation of AMPK depends on the severity of nutrient or energy stress [48]. In the present study, we further observed that AMPK phosphorylation was mildly suppressed in the early stage of high-glucose-induced vascular injury in vivo, suggesting that hierarchical signaling of AMPK depends on the severity and exposure of nutrient or energy stress. Many researches have confirmed the beneficial effects of AMPK activation on diabetic or high-glucose-induced cellular apoptosis and endothelial dysfunction [49]. For instance, Empagliflozin rescues diabetic myocardial microvascular injury via AMPK activation [50]. The role of AMPK activation as a valuable strategy for rebalancing mitochondrial fusion and fission in the progression of high-glucose-induced vascular injury is proposed. However, our results show that DOR does not inhibit mitochondrial fusion proteins as high-glucose exposure does. The effect of AMPK inhibition on mitochondrial fusion remains to be further studied.

Although we have found the staged alteration of mitochondrial fusion/fission during high-glucose-induced vascular injury, the details of crosstalk between mitochondrial fusion/fission proteins at different balance stages remain unknown. Studying the optimal treatments at different stages will help manage diabetic vascular complications.

## 5. Conclusions

In conclusion, our findings uncover that shifting mitochondrial fusion/fission homeostasis drives the high-glucose-induced vascular injury, which is rapidly developed in response to high glucose. For this reason, the staged homeostasis of mitochondrial fusion/fission may be a predictive factor of high-glucose-induced vascular injury. It will be a promising strategy to restore the mitochondrial fusion/fission homeostasis via fusion activation instead of fission inhibition.

## Figures and Tables

**Figure 1 biomolecules-11-01779-f001:**
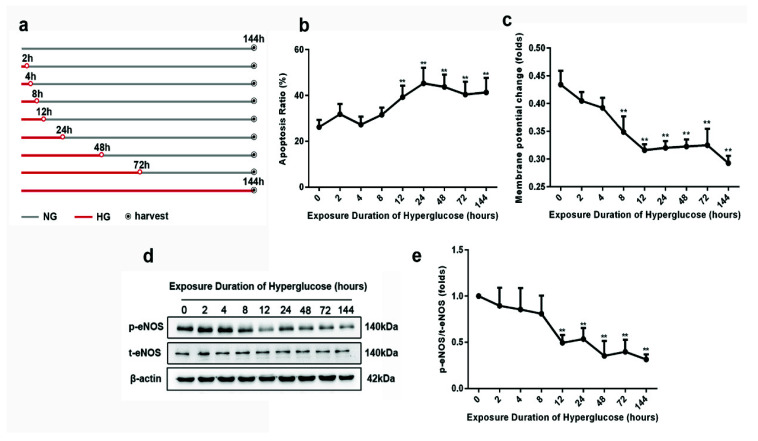
Metabolic memory formation in HUVECs after different high-glucose exposure times. (**a**) Scheme of cell culture. (**b**) Apoptosis assay. (**c**) Depolarization of mitochondrial membrane potential (MMP). (**d**) The protein expression of phosphorylated eNOS and total eNOS. (**e**) Quantitative analysis of eNOS phosphorylation, obtained by normalization based on the ratio of phosphorated-eNOS expression to total-eNOS expression. Data are expressed as the mean ± SEM. *n* = 3 for each group. ** *p* < 0.01 vs. the normal-glucose group.

**Figure 2 biomolecules-11-01779-f002:**
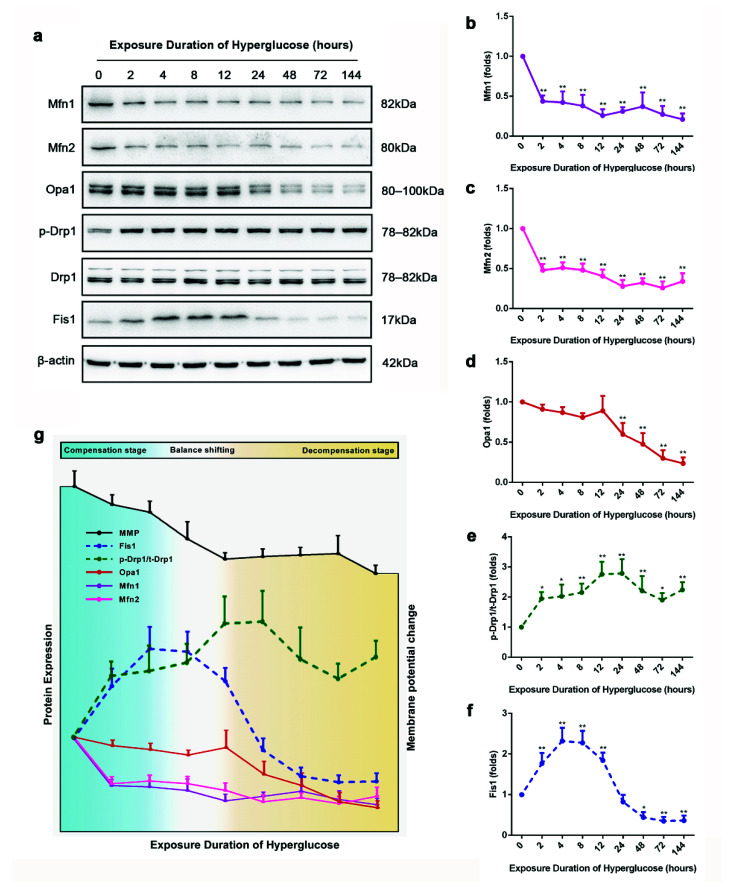
The three stages of mitochondrial fusion/fission balance during high-glucose-induced vascular injury. (**a**–**f**) Expression levels of mitochondrial fusion/fission proteins. (**g**) Summary of the sustained effects of different high-glucose exposure times on MMP, expression of mitochondrial dynamic proteins, and cellular apoptosis assay. Data are expressed as the mean ± SEM. *n* = 3 for each group. * *p* < 0.05, ** *p* < 0.01 vs. the normal-glucose group.

**Figure 3 biomolecules-11-01779-f003:**
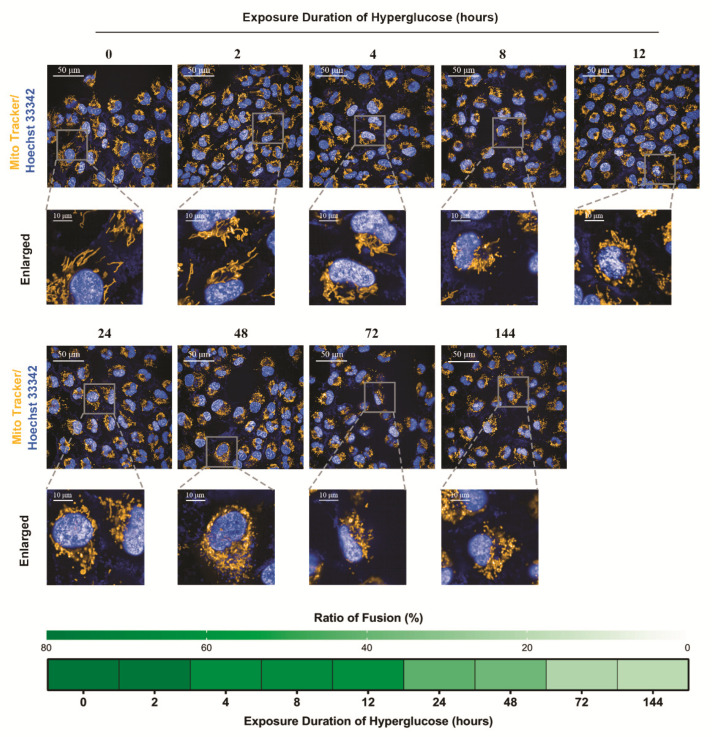
Sustained effects of different high-glucose exposure times on mitochondrial morphology. The status of mitochondrial integrity is shown as the ratio of fusion.

**Figure 4 biomolecules-11-01779-f004:**
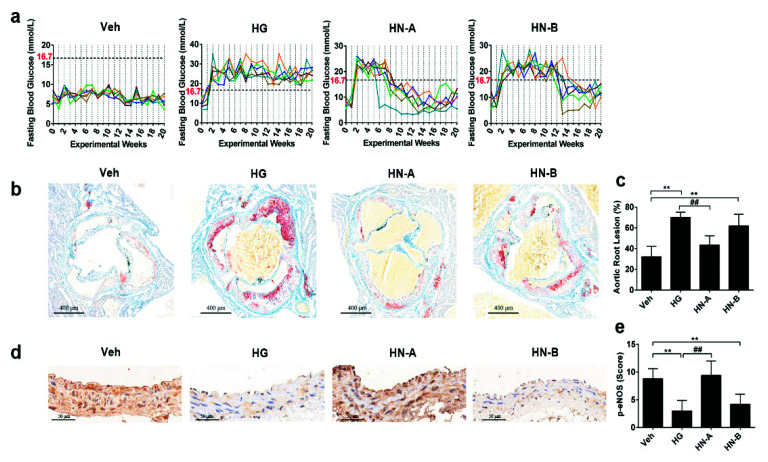
High-glucose-induced vascular injury in vivo. (**a**) The fasting blood glucose of mice. (**b**,**c**) Oil red staining and quantitative analysis of atherosclerotic lesions at the aortic root. (**d**,**e**) Immunohistochemical staining and quantification of p-eNOS on abdominal aortic sections. Data are expressed as the mean ± SEM. *n* = 6 for each group. ** *p* < 0.01 vs. control (the Veh group); ## *p* < 0.01 vs. HG.

**Figure 5 biomolecules-11-01779-f005:**
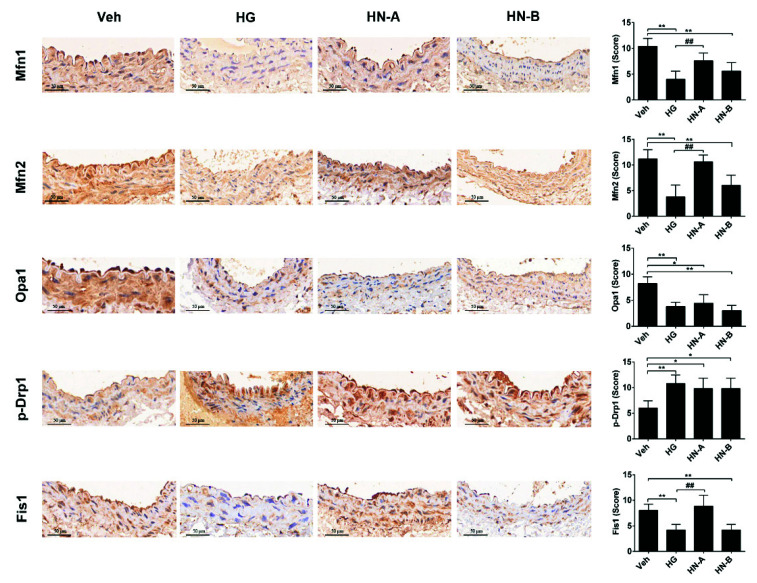
Immunohistochemical staining and quantification of mitochondrial fusion/fission proteins in abdominal aortic sections. Data are expressed as the mean ± SEM. *n* = 6 for each group. * *p* < 0.05, ** *p* < 0.01 vs. control (the Veh group); ## *p* < 0.01 vs. HG.

**Figure 6 biomolecules-11-01779-f006:**
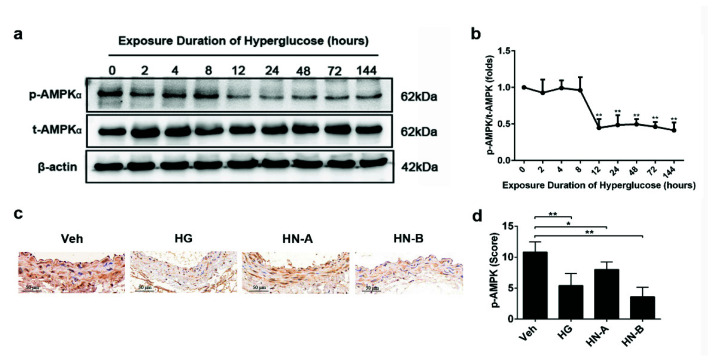
Adenosine monophosphate-activated kinase (AMPK) phosphorylation was suppressed during high-glucose-induced vascular injury both in vitro and in vivo. (**a**,**b**) AMPK phosphorylation in HUVECs with different high-glucose exposure times. Data are expressed as the mean ± SEM. *n* = 3 for each group. * *p* < 0.05, ** *p* < 0.01 vs. normal-glucose group (the 0 h group). (**c**,**d**) Immunohistochemical staining and quantification of p-eNOS on abdominal aortic sections. Data are expressed as the mean ± SEM. *n* = 6 for each group. * *p* < 0.05, ** *p* < 0.01 vs. control (the Veh group).

**Figure 7 biomolecules-11-01779-f007:**
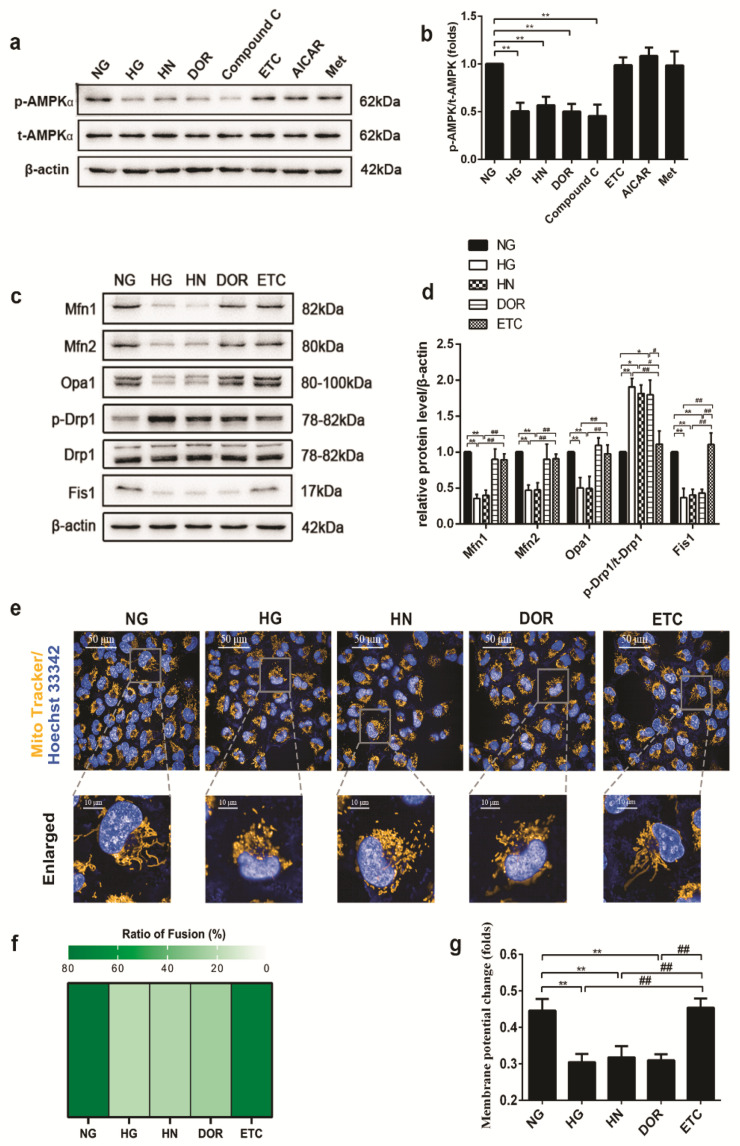
AMPK phosphorylation mediates the mitochondrial fusion/fission balance shifting in HUVECs. (**a**,**b**) The effects of some classical AMPK activators and inhibitors on AMPK phosphorylation. (**c**,**d**) Effects of DOR, ETC on the mitochondrial fusion/fission proteins. (**e**,**f**) Effects of DOR, ETC on the mitochondrial morphology. The status of mitochondrial integrity is shown as the ratio of fusion. (**g**) Effects of DOR, ETC on the mitochondrial membrane potential. NG, the negative control; HG, the positive control; HN, the model control; ETC, Compound C, Met, the AMPK agonists; DOR, AICAR, the AMPK inhibitor. Data are expressed as the mean ± SEM. *n* = 3 for each group. * *p* < 0.05, ** *p* < 0.01 vs. control (the NG group); # *p* < 0.05, ## *p* < 0.01, vs. ETC.

## Data Availability

Data are available on request.

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
