# Peer review of "The Imbalance of Mitochondrial Fusion/Fission Drives High-Glucose-Induced Vascular Injury"

_biomolecules, 2021, doi:10.3390/biom11121779_

Round 1

Reviewer 1 Report

The authors demonstrated that HUVECs exposed to high glucose exhibit an imbalance in mitochondrial fusion / fission, with the presence of fragmented mitochondria, which in turn contribute to impaired endothelial function and apoptosis. The in vitro data were corroborated in ApoE mice undergoing treatment with streptozocin. At the molecular level, AMPK decreases under hyperglycemic conditions and AMPK activation rescues mitochondrial dynamics.

The work is very interesting and results are convincing. I have some comments that should be addressed:

1) Authors should demonstrate that AMPK activation also rescues apoptosis and endothelial dysfunction in high glucose / diabetic models.

2) Observing mitochondrial dynamics, mitochondrial fission appears to increase during prolonged high-glucose treatment. Is inhibition of mitochondrial dynamics able to decrease apoptosis?

3) The scale bar should be added to all microscopy images

Author Response

Responses to reviewer #1:

  1. To comment, “Authors should demonstrate that AMPK activation also rescues apoptosis and endothelial dysfunction in high glucose/diabetic models.”

Response: Thank you for your comment. We have added this issue in the discussion section (Page 15, lines 400 to 403). In this study, our major concern is the role of mitochondrial fusion and fission balance shifting in high-glucose-induced vascular injury. The beneficial effects of AMPK activation on diabetic or high-glucose-induced cellular apoptosis and endothelial dysfunction have been demonstrated (Steinberg 2018 in DIABETES). In this study, AMPK activator bempedoic acid was used to demonstrate that AMPK mediate mitochondrial fusion and fission balance shifting.

  1. To comment, “Observing mitochondrial dynamics, mitochondrial fission appears to increase during prolonged high-glucose treatment. Is inhibition of mitochondrial dynamics able to decrease apoptosis?”

Response: It is a good question. Fragmented mitochondria arise from inhibited fusion or accelerated fission. Mitochondrial dynamics inhibition can be achieved by activation of fusion or inhibition of fission. Activation of fusion can be achieved by upregulating Mfn1, Mfn2 and Opa1, while inhibition of fission can be achieved by suppressing Drp1 or Fis1. In another research, we compare several different methods of restoring mitochondrial dynamics. We find that inhibition of Drp1 by Midiv-1 can rescue apoptosis and endothelial dysfunction in the high glucose model within a special time (Unaccomplished data).

  1. To comment, “The scale bar should be added to all microscopy images.”

Response: We feel sorry for our sloppiness and thank you again for the careful check. We have done these modifications in Figure. 3, 4, 5, 6, and 7.

Reviewer 2 Report

Manuscript " The Imbalance of Mitochondrial Fusion/Fission Drives High-Glucose-Induced Vascular Injury" elucidates the role of mitochondrial fusion/fission in vascular injury in a high glucose-dependent manner. this manuscript is suitable for publication if following question should be addresed.

  1. Line number 182; the author should explain the function of eNOS protein and its regulation in hyperglycemic conditions and also detailed information of protein in the introduction section.
  2. Figure 1 (e). densitometry of immunoblot is not representative of the result. 
  3. The author should provide the method for analysis of fusion (figure 3)
  4. The author should explain what is STZ (line number 233) and its function.
  5. The End 

Author Response

Responses to reviewer #2:

  1. To comment “Line number 182; the author should explain the function of eNOS protein and its regulation in hyperglycemic conditions and also detailed information of protein in the introduction section.”

Response: Thank you for the valuable comment. We have added these contents in the introduction section and make it clear (Page 2, lines 61 to 68).

  1. To comment “Figure 1 (e). densitometry of immunoblot is not representative of the result.

Response: We appreciate your comment. We are very sorry that we did not clearly describe the data processing method. Figure 1 (e) is obtained by normalization based on the ratio of phosphorated-eNOS expression to total-eNOS expression. To avoid misunderstanding the results, we have added explanations in the figure legend (Page 5, lines 203 to 205).

  1. To comment “The author should provide the method for analysis of fusion (figure 3).

Response: We regret that we did not make it clear. We have revised it in the methods section (Page 4, lines 177 to 182). For analysis of the ratio of fusion, the following steps were carried out in the Harmony® software: the nucleus and mitochondrial skeleton were identified; then the mitochondria and cells were mapped; the mitochondrial length of the normal-glucose group was adjusted so that the fragmented mitochondria accounted for about 20%; this mitochondrial length was set as the standard to distinguish fusion and fission, the fusion ratios of other groups were obtained.

  1. To comment “The author should explain what is STZ (line number 233) and its function.

Response: Thank you for your comment. We have added this content in the revised manuscript (Page 8, lines 249 to 250). STZ (Streptozotocin) selectively destroys the islet cells and induces hyperglycemia. We used STZ to produce a diabetic model of depleted insulin secretion.

       5.The end.

Round 2

Reviewer 1 Report

No further comments